# 1,2,5-Thiadiazole 1,1-dioxides and Their Radical Anions: Structure, Properties, Reactivity, and Potential Use in the Construction of Functional Molecular Materials

**DOI:** 10.3390/molecules26164873

**Published:** 2021-08-11

**Authors:** Paweł Pakulski, Dawid Pinkowicz

**Affiliations:** Faculty of Chemistry, Jagiellonian University, Gronostajowa 2, 30-387 Kraków, Poland

**Keywords:** 1,2,5-thiadiazole 1,1-dioxides, molecular materials, persistent organic radicals, molecular magnetism

## Abstract

This work provides a summary of the preparation, structure, reactivity, physicochemical properties, and main uses of 1,2,5-thiadiazole 1,1-dioxides in chemistry and material sciences. An overview of all currently known structures containing the 1,2,5-thiadiazole 1,1-dioxide motif (including the anions radical species) is provided according to the Cambridge Structural Database search. The analysis of the bond lengths typical for neutral and anion radical species is performed, providing a useful tool for unambiguous assessment of the valence state of the dioxothiadiazole-based compounds based solely on the structural data. Theoretical methodologies used in the literature to describe the dioxothiadiazoles are also shortly discussed, together with the typical ‘fingerprint’ of the dioxothiadiazole ring reported by means of various spectroscopic techniques (NMR, IR, UV-Vis). The second part describes the synthetic strategies leading to 1,2,5-thiadiazole 1,1-dioxides followed by the discussion of their electrochemistry and reactivity including mainly the chemical methods for the successful reduction of dioxothiadiazoles to their anion radical forms and the ability to form coordination compounds. Finally, the magnetic properties of dioxothiadiazole radical anions and the metal complexes involving dioxothiadiazoles as ligands are discussed, including simple alkali metal salts and *d*-block coordination compounds. The last section is a prospect of other uses of dioxothiadiazole-containing molecules reported in the literature followed by the perspectives and possible future research directions involving these compounds.

## 1. Introduction

Organic molecules are at the very edge of modern applications in technology (OLEDS, organic conducting materials, batteries) and there is a never-ending need for new systems that meet more and more demanding requirements. Heterocyclic organic compounds containing both N and S atoms are widely studied throughout the chemical and molecular sciences in this regard. Specifically, thiadiazoles [1,2,3,4,5,6,7,8,9] (Figure 1a) show several advantageous features such as high thermal and chemical stability, easy synthetic preparation and availability of precursors, coordination and bonding abilities [10], rich electrochemistry [11,12,13], and other functionalities such as magnetism [14], luminescence [15,16,17,18], or chirality [19]. The thiadiazole ring provides a very convenient tool for tuning the physicochemical properties of the relevant molecules—both nitrogen and sulfur atoms can be oxidized with common laboratory oxidants to yield *N*-oxides, *S*-oxides (oxothiadiazoles), and *S*,*S*-dioxides (dioxothiadiazoles).

Surprisingly, despite great interest in thiadiazole chemistry, little is done to harness the potential of the oxidized species [20]. Dioxothiadiazoles, in this context, are underrepresented, and their chemistry is still awaiting to be recognized as, at the very least, equally interesting as the parent thiadiazoles. The oxo- and dioxothiadiazoles can be easily reduced to stable radical anions [21] and are therefore of potential use to the molecular magnetism community. However, this group of compounds seems to be completely omitted as potential building blocks for the construction of molecule-based magnetic materials [22,23] compared to other organic radicals (nitronyl nitroxides [24,25,26,27], verdazyl [28,29,30], semiquinones [31,32], or TCNE and TCNQ derivatives [33,34]) which have been used extensively to obtain all-organic magnets [35], Single Molecule Magnets [36,37], or Single Chain Magnets (SCMs) [24,38,39,40,41], among many other functional magnetic materials. This review focuses on 1,2,5-thiadiazole 1,1-dioxides (Figure 1a) and their radical anions [42], which constitute perhaps the prime example of the dioxothiadiazole family, and summarizes the current state-of-the-art in the field of 1,2,5-thiadiazole 1,1-dioxides with the aim of shedding light on the extraordinary possibilities and applications of this heterocyclic moieties with respect to the design and preparation of switchable/functional molecular materials.

## 2. Structure and Geometry

1,2,5-thiadiazole 1,1-dioxides (hereafter denoted as dioxothiadiazoles) comprise a five-membered heterocyclic ring oxidized at the sulfur atom. The ring is attached via the carbon atoms to various organic “backbones” as depicted in Figure 1b (please note that Figure 1b gathers all crystallographically characterized 1,2,5-thiadiazole 1,1-dioxides excluding thiadiazolines and the related N-functionalized derivatives).

Table 1 supplements Figure 1b by providing references to the CSD database and the relevant publications, where the respective compounds are reported [43,44,45,46,47,48,49,50,51]. Sulfonyl (>SO_2_) is a very strong electron-withdrawing group; therefore, compared to thiadiazoles, their dioxidized analogues containing this sulfonyl group are superior when it comes to negative charge accommodation within the heterocyclic ring and the resulting radical anion formation and their stability. In fact, most of the dioxothiadiazole species can be successfully reduced not only to their monoanionic radical forms but also to the dianionic species, while regular thiadiazoles are not prone to be reduced twice, and the first reduction event occurs usually around −2.0 V vs. Fc/Fc^+^ [42]. The sulfonyl group is also modifying the nature of molecular interactions available for the heterocyclic ring when transitioning from thiadiazoles to dioxothiadiazoles: the exposed soft sulfur atom in thiadiazoles, which is useful for the coordination to soft metal ions such as Cu^+^, Ag^+^ and Hg^2+^ as well as the weak hydrogen bonding capabilities, is replaced by hard oxygen atoms in dioxothiadiazoles that prefer hard metal ions such as alkali metals and can easily form hydrogen bonds.

A detailed structural analysis of the dioxothiadiazole heterocycle in various derivatives, as presented in Table 2 and Figure 2, indicates that the respective bonds within the heterocycle change length in a systematic way when the compound is reduced to the radical anion form.

Figure 2 presents the histograms of all bond lengths within the dioxothiadiazole ring gathered in Table 2 in their neutral and radical forms (0.005 Å intervals on the bond length axis). The largest bond length change can be observed for the C-C and C=N bonds, which can be used as markers of the valence state of the dioxothiadiazole derivative. This may be particularly useful when dioxothiadiazoles are used to form complexes with redox-active metal centers to form organic–inorganic molecular frameworks with valence tautomerism—with just a crystal structure, it is possible to accurately determine the valence state of the ligands. Upon reduction, the C-C and S-N bonds get shorter, while C=N and S=O bonds get longer.

The first systematic investigation of the properties of dioxothiadiazoles and perhaps the first rational approach to this heterocycle as a promising candidate for the design of molecular materials was carried out by Awaga et al. His group focused on dioxothiadiazoles fused to extended aromatic systems such as 1,10-phenanthroline (1,10-tdapO_2_) or pincene and performed a general physicochemical evaluation of these systems [14,44,51,52]. [1,2,5]thiadiazolo[3,4-f][1,10]phenanthroline 2,2-dioxide (1,10-tdapO_2_) structurally embodies all functionalities necessary to use this building block in the construction of mixed organic–inorganic coordination systems [53]. It readily forms radical anions with stability enhanced by the delocalization of the spin density over a large aromatic system of the 1,10-phenanthroline moiety. This can be clearly seen by comparing 1,10-tdapO_2_ with monocyclic analogues of dioxothiadiazoles such as 3,4-diphenyl-1,2,5-thiadiazole 1,1-dioxide. Moreover, the nitrogen atoms of its 1,10-phenanthroline backbone are capable of forming stable complexes with *d*-block metal ions, while the oxygen atoms of the dioxothiadiazole group can form electrostatic interactions with alkali metal ions. All heteroatoms of 1,10-tdapO_2_ may also participate in hydrogen bonding, providing directionality and enhancing the rigidity of the molecular frameworks formed. Additionally, the flat π electron-rich surface of the tdapO_2_ can form π-π stacking interactions that controls to some extent the ordering of molecules within the crystals. When distances between the 1,10-tdapO_2_ radical species are small and the offset appropriate, these stacks become important magnetic interaction pathways [14,44,52]. Please see the following sections for more details.

### 2.1. Theoretical Investigations

Several theoretical works with a detailed description of selected dioxothiadiazole molecules have been published, providing some general insight into the computational methods best suited to yield satisfactory agreement with experimental data for this family of compounds [47,50,54,55,56,57,58]. The most popular methods for the calculation of dioxothiadiazole properties is standard and involve DFT with B3LYP hybrid functional and a 6-311G++(d,p) or 6-311G++(3df,3pd) basis set. As in the case of most organic molecules, it is fast and reliable in reproducing data for UV-vis and IR interpretation that closely resemble experimental results [21,42,45,51,59,60]. For EPR spectra simulations, usually a B98 hybrid functional is used instead. In the case of solvents effects, the Self-Consistent Isodensity Polarized Continuum Model (SCI-PCM) is employed [59]. CHIH-DFT (Chihuahua Heterocycles-Density Functional Theory) model chemistry is also proposed as a semi-empirical DFT method specifically tailored to accurately predict properties of heterocyclic systems such as dioxothiadiazoles [61].

### 2.2. NMR, UV-Vis, and IR Spectroscopy

Simply because there are no hydrogen atoms present within the 1,2,5-thiadiazole 1,1-dioxide moiety, the ^1^H NMR spectra are useless when it comes to the structural characterization of this class of compounds. Several ^13^C NMR spectra have been reported for 3,4-disubstituted derivatives; carbon signals originating from 1,2,5-thiadiazole 1,1-dioxide ring appear in the 150–170 ppm range [59,60,62,63,64,65,66,67,68] as compared to the 130–160 ppm range for 1,2,5-thiadiazoles [69,70,71]. Many compounds were not characterized by ^13^C NMR spectroscopy due to their relatively low solubility in common deuterated solvents [43,45,51,67].

IR spectroscopy is quite useful for the characterization/identification of dioxothiadiazoles. This technique can be easily performed in the solid state, which mitigates their poor solubility. Four distinct IR bands are attributed to the heterocyclic ring: two C=N stretching vibrations in the 1600–1550 cm^−1^ range and two S=O stretches at 1350–1280 and 1170–1100 cm^−^^1^ [20,63,64] enable identification of dioxothiadiazole derivatives in more complex supramolecular and coordination systems. Figure 3 presents IR spectra (fingerprint region) of two exemplary dioxothiadiazole derivatives with a clear indication of the C=N and S=O stretching vibration bands.

UV-Vis spectroscopy for the simplest 3,4-disubstituted 1,2,5-thiadiazole 1,1-dioxides has a maximum in the ultraviolet region (approximately 315 nm in MeCN and 240–280 nm in ethanol) with the molar absorption coefficient on the order of 10^3^ dm^3^·mol^−1^·cm^−1^ [68,72,73]. When the heterocyclic ring is fused to a larger aromatic system, the UV-Vis spectra become complicated, and the ε_max_ often exceeds 10^4^ dm^3^·mol^−1^·cm^−1^ in the UV region [56,60,68,74].

## 3. Preparation Methods

Synthetic strategies toward 1,2,5-thiadiazole 1,1-dioxides are scarce and can be divided into two distinct categories: condensation of diketones with sulfamide [63,64,72,74,75,76,77] or oxidations of the pre-existing thiadiazoles or 1,2,5-thiadiazole oxides to the corresponding dioxides (Figure 4, Table 3) [21,67,78,79].

Out of the four possible synthetic routes leading to dioxothiadiazoles, the simplest and the most versatile one is the condensation reaction between sulfamide and 1,2-diketones (α-diketones) [75], as clearly indicated by Table 2. Another possible substrate for condensations with sulfamide is cyanogen [63]. However, this reaction leads to 1,2,5-thiadiazole 1,1-dioxides as the only possible derivative, which could be further functionalized by C-C coupling reactions. Small-scale preparations can resort to other substrates for condensations originating from diketone scaffolds such as methyl 2-oxo-2-phenylacetate and its analogues as well as dimethyl or diethyl oxalate [72,76].

Condensations with sulfamide can also benefit from several modifications in order to deal with common problems such as the presence of strong mineral acid and prolonged heating at elevated temperature or the fact that sulfamide is not readily soluble in aprotic solvents. Poor sulfamide solubility in aprotic media can be evaded by the use of its substituted analogue-*N,N’*-bis(trimethylsilyl)sulfamide prepared from SO_2_Cl_2_ and NH(SiMe_3_)_2_. To remove the silyl groups from the product and force the reaction to proceed, a stoichiometric amount of BF_3_·Et_2_O complex is required. This synthetic procedure does not involve strong acid catalyst and is performed efficiently at room temperature, thus providing an excellent alternative for heat-sensitive substrates with functional groups incompatible with strong mineral acids [64].

Another alternative for strong mineral acids may be the use of heteropolyacids (HPAs), more precisely, a commercially available and cheap Keggin-type molybdophosphoric acid H_3_PMo_12_O_40_·nH_2_O [74]. Reactions carried out with this catalyst can be performed in a solvent-free fashion, making them an attractive, green, and environment-friendly alternative to the conventional solvent-based approaches. Efforts are being made to develop a silica-supported version of this catalyst to further decrease the catalyst load required for the efficient dioxothiadiazole synthesis. Microwave-assisted synthesis has also been proposed and successfully applied in dioxothiadiazole base-mediated preparations [77].

When it comes to oxidations of pre-existing thiadiazoles and oxothiadiazoles to the respective dioxides, there is only one exclusive route involving mCPBA reported in the literature [21,78]. There is also a possibility of transforming 1,2,5-dioxothiadiazolines back to dioxothiadiazoles with the use of KO_2_ in THF or NaOEt in DMF [67,79].

## 4. Reactivity

### 4.1. Reactivity toward Nucleophiles

Although the dioxothiadiazole ring is stable enough for sublimation at temperatures up to 300 °C, elevated temperature can decompose it, liberating SO_2_ and leaving two nitrile groups or a thiadiazole ring. This reaction was successfully applied in gram-scale preparations of dinitriles [65,80,81].

The 1,2,5-dioxothiadiazole ring is especially vulnerable to nucleophiles, and additional reactions of various nucleophiles to the C=N double bond are described. In fact, this reaction is so favorable, it occurs spontaneously in EtOH or EtOH/MeCN solutions of 3,4-diphenyl- or 3-methyl-4-phenyl-1,2,5-thiadiazole 1,1-dioxide [82,83,84]. This has proven to be a general reactivity of 3,4-substituted-1,2,5-thiadiazole 1,1-dioxides toward alcohols; many examples of addition of primary and secondary alcohols were presented and described in details, providing equilibrium constants for these reactions [68]. The same equilibrium and addition to double C=N bond can be observed for primary/secondary amines and amides; see Figure 5 [85,86].

Products of such additions are not stable and exist only in solution. However, for some nucleophiles, this reversible addition to both C=N double bonds may lead even to an irreversible ring cleavage with the release of sulfamide molecules when the reaction is carried out in anhydrous MeCN or DMF at room temperature; see Figure 6 [85,87]. Double additions can also be achieved using urea (or thiourea) and its substituted analogues to furnish the bicyclic products presented in Figure 7 [73,86].

Cyanide ions are capable of mono or double addition to the 1,2,5-dioxothiadiazole 1,1-dioxide ring at positions 3 and 4 depending on the molar ratio of the heterocycle and the nucleophile. Unstable adducts can be recovered by the addition of methyl iodide to capture substituted N-methylated thiadiazoline and thiadiazolidine oxides, respectively [88]. The addition of alkyl halides to the solution of 3,4-diphenyl-1,2,5-thiadiazole 1,1-dioxide in methanol and subsequent treatment with thiourea in acidic media yields N-alkylated thiadiazoline oxides [89].

The C=N double bond can also be functionalized with alkyl and aryl Grignard reagents. Although the addition proceeds smoothly with no side reactions, products are reported to be unstable in the chromatographic purification (silica gel) and must be used as crudes for further reactions such as reduction with sodium borohydride to achieve thiadiazolidine dioxides [90].

An AlCl_3_–catalyzed addition of aromatic nucleophiles to the C=N bond of 3,4-diphenyl 1,2,5-thiadiazole was successfully carried out in DCM at room temperature yielding substituted 1,2,5-thiadiazolines in moderate to good yields (Figure 8) [46,91].

When chiral reagents based on enantiopure binol and rhodium catalysts are used together with vinylboronic acids, this addition to the dioxothiadiazole heterocycle can be performed in a stereo-controlled manner to yield optically active 1,2,5-thiadiazoline 1,1-dioxides with 90+% yields and ee exceeding 90% [76].

Aromatic rings as substituents at positions 3 and 4 can undergo intramolecular cyclization/aromatization (Scholl reaction) catalyzed by strong Lewis (AlCl_3_) or Brønsted (HClSO_3_/H_2_SO_4_) acids. However, this reaction suffers from some limitations regarding the substrates and was only successfully applied in two cases: 3,4-diphenyl 1,2,5-thiadiazole 1,1-dioxide and 3,4-di(naphthalen-2-yl)-1,2,5-thiadiazole 1,1-dioxide [92].

As a result of the strong electron-withdrawing nature of the sulfonyl group, any alpha hydrogens at carbon positions 3 and 4 are very acidic and can easily be removed in basic environment to form tautomeric thiadiazoline carbanions that can be isolated by the addition of strong acids such as TFA (Figure 9) [93].

The reduction of dioxothiadiazoles to dioxothiadiazolidines proceeds smoothly with the use of hydrogen and Adams’ catalysts (PtO_2_) [75]. Double bonds of 1,2,5-thiadiazole dioxide can also be oxidized to furnish fused bis-oxaziridine derivatives (Figure 10) [3].

The abstraction of two oxygen atoms and reduction to parent 1,2,5-thiadiazole was performed for [1,2,5]thiadiazolo[3,4-f]-4,7phenanthroline 2,2-dioxide through extensive heating (285 °C) in vacuum for a prolonged time (15 h) in a sealed Pyrex tube with a partial decomposition of the initial compound. Product was collected via sublimation [43].

### 4.2. Radical Anion Formation and Electrochemistry

One of the most researched and important features of dioxothiadiazoles is their rich and interesting electrochemistry, which is heavily influenced by the substitution of the heterocyclic ring at the carbon atoms. Several methods have been developed to produce radical and dianionic species and investigate how structural changes influence the electrochemical behavior of dioxothiadiazoles.

Anion radicals can be generated by electrochemical methods or with the use of common laboratory reducing reagents including CN^−^, SCN^−^, OCN^−^, I^−^, OH^−^, t-BuO^−^, amines (triethylamine, ethanolamine, hexamethylenediamine, and N,N-dimethylaminoethylenediamine), and even amides (formamide, N-methylformamide DMF, acetamide, or urea) [42,59]. Electrochemical reduction performed in a controlled potential electrolysis fashion can yield almost quantitative amounts of radicals, with net charge close to one Faraday per mole, indicating that almost no side processes are taking place. In aprotic solvents (DMF, DMSO, MeCN, or DCM) some of the radicals are stable for prolonged periods of time (up to several days), even in the presence of water and oxygen, in dry solvents, they are stable almost indefinitely. The stability of the radical anions depends on the solvent used and decreases in the following series: DMF > DMSO > MeCN >> DCM [48]. It is noteworthy that the use of highly nucleophilic reductors, such as cyanide, may result in the formation of side products by addition to the C=N double bond of the heterocycle. Therefore, sterically hindered reducing agents are preferred, as they allow for better selectivity of anion radical formation; however, even the change of the cation in the system (e.g., KCN vs. LiCN) may play an important role in avoiding the addition reaction [42]. Finally, radical anions of dioxothiadiazoles can be generated with moderate yields in the photochemical approach using the wavelength of around 254 nm and DMF as a solvent [48]. However, reduction is taking place because of the initial radical formation involving DMF molecules that further reduce dioxothiadiazole species. This might be followed by side reactions originating from the photodegradation of DMF which makes the photochemical radical formation unappealing for preparative or practical use.

Typically, two distinct, reversible reduction events can be observed for any dioxothiadiazole within the MeCN electrochemical window. Reduction to radical anions usually takes place at around −0.5 V vs. Fc/Fc^+^ and is followed by the second reduction to a diamagnetic dianion at around −1.3 V [14,21,43,44,48,51,60,94]. It is noteworthy that most thiadiazoles cannot be reduced twice at all and are hardly even reduced to monoanions [21,43,44,94,95]. Electrochemistry, HOMO, and LUMO levels can be easily manipulated and tuned by the introduction of electron-withdrawing substituents such as bromine atoms [45,48].

## 5. Magnetic Properties of Dioxothiadiazole Radical Anions

Due to the exceptional stability of the dioxothiadiazole radical species, they can be studied using standard techniques such as electron paramagnetic resonance (EPR) or magnetometry. The same applies to their compounds: metal salts and coordination complexes. The following section will summarize and discuss all relevant radical anion-based magnetic materials.

### 5.1. Electron Paramagnetic Resonance

There are several reports providing some insights into the stability and structure of the anion radical species based on the 1,1-dioxo-1,2,5-thiadiazole motif [42,48,59]. In general, *g* factors for radicals based on dioxothiadiazoles vary within the range of 2.002–2.009. This indicates that the unpaired electron responsible for its formation is delocalized. Hyperfine coupling constants for nitrogen atoms of the 1,1-dioxo-1,2,5-thiadiazole ring *a_N_* of around 2.5–4 G are usually present in EPR spectra of these anion radicals [14,42,43,51,59]. The presence of the -NO_2_ group in the structure of the radical backbone, even if it is seemingly distant from the dioxothiadiazole ring, is enough to draw a major part (about 2/3) of the electron spin density toward it, affecting the *g* factor value and EPR spectra because of stronger interaction with nitrogen atom reflected in strong (ca. 11 G) hyperfine coupling [59].

### 5.2. Magnetometry

Having an unpaired electron, anion radicals based on 1,2,5-dioxothiadiazole 1,1-dioxides are paramagnetic. Therefore, they can interact magnetically with other spin carriers in the solid state, leading to peculiar magnetic behaviors, including the long-range ferro-/antiferromagnetic ordering. This line of research is well known in the field of molecular magnetism with many fascinating examples [24,35,36,37,38,39,40,41]. Dioxothiadiazoles have drawn little attention in this regard, and only recently, some interesting results have been published [43,44,45,52,53].

Nevertheless, all magnetic systems published so far involve dioxothiadiazoles fused to a larger π-electronic system such as phenanthroline [14,43,44,45,52] or pincene [51]. Most commonly, those systems present antiferromagnetic superexchange coupling with *J*/*k*_B_ values between −13 and −310 K (for the spin Hamiltonian *H = −2 J**ΣS_i_S_i+1_*), arising from the magnetic interactions between two π-π stacked dioxothiadiazoles [14,44,45,51]. Figure 11 presents an exemplary *χT*(*T*) dependence for π-π stacked structure of K^+^[4,7-tdapO_2_]^−^ [43] due to the very strong antiferromagnetic interactions between the radical anions. Ferromagnetic interactions are also observed with *J*/*k*_B_ value in the 6–24 K range [14]. In light of this information, it becomes clear that for this type of system, stacking interactions are crucial for the successful design of molecule-based magnets with desired magnetic properties.

## 6. Magnetic Materials with Dioxothiadiazoles as Ligands

Two significantly different classes of compounds can be distinguished among examples of dioxothiadiazole-based magnetic materials: organic and alkali metal salts of dioxothiadiazole anion radicals [14,43,44,51] and *d*-block metal ions coordinated by the dioxothiadiazoles [43,53]. Most of the coordination abilities of the dioxothiadiazole heterocycle are realized through its oxygen atoms, although there are examples where nitrogen atoms are also involved in coordination to metal centers. There are numerous examples of alkali metal salts of dioxothiadiazole anions where K^+^, Rb^+^, or Cs^+^ cations are interacting with the oxygen atoms of the dioxothiadiazole moiety and a few examples where nitrogen atoms are utilized together with the oxygen atoms to coordinate to metal centers [14]. There is also one example of a coordination chain {CuCl_2_(4,7-tdapO_2_)}_n_ (4,7-tdapO_2_ = [1,2,5]thiadiazolo[3,4-f][4,7]phenanthroline 2,2-dioxide) where copper ions are coordinated by one nitrogen of the dioxothiadiazole unit and one nitrogen atom of the 4,7-phenanthroline backbone and the diamagnetic 4,7-tdapO_2_ ligand acts as a molecular bridge (Figure 12) [43]. Magnetic measurements for {CuCl_2_(4,7-tdapO_2_)}_n_ revealed that it behaves as a magnetic chain with Cu^II^⋅⋅⋅Cu^II^ antiferromagnetic interactions transmitted through the bridging ligand, dominating the magnetic properties of the compound. Dioxothiadiazole moiety can contribute to the functionality of the designed coordination complexes by utilizing its rich electrochemistry to tune the properties of the material. An example of such use of dioxothiadiazoles is a series of complexes of [1,2,5]thiadiazolo[3,4-f][1,10]phenanthroline 2,2-dioxide (1,10-tdapO_2_) with Cu^II^ ions where the metal was coordinated by two 1,10-tdapO_2_ moieties with different valence states: solely neutral 1,10-tdapO_2_ in {[Cu^II^Cl(1,10-tdapO_2_)](μ-Cl)_2_[Cu^II^Cl(1,10-tdapO_2_)]}, mixed neutral and radical 1,10-tdapO_2_ in [Cu^II^Cl(1,10-tdapO_2_^−^)(1,10-tdapO_2_)]·2MeCN and solely radical 1,10-tdapO_2_^−^ in PPN[Cu^II^Cl(1,10-tdapO_2_^−^)_2_]·2DMA (DMA = dimethylacetamide; PPN = bis (triphenylphosphine) iminium cation); see Figure 13.

An interesting point was made about the redox potentials of the 1,10-tdapO_2_ molecule changing upon the coordination to the positively charged, electron-withdrawing metal ion. This change was found to be about +300 mV as compared to the free ligand with the first reduction potential at around −520 mV vs. Fc/Fc^+^. In terms of the magnetic properties, the three reported compounds are paramagnetic and show antiferromagnetic interactions, which are revealed as the decrease of the *χT*(*T*) product while cooling. In the case of the system with mixed neutral and radical 1,10-tdapO_2_ ligands, [Cu^II^Cl(1,10-tdapO_2_^·−^)(1,10-tdapO_2_)]·2MeCN, its magnetic properties are dominated by strong antiferromagnetic interaction between the neighboring 1,10-tdapO_2_ radical anions rather than the intramolecular 1,10-tdapO_2_-Cu^II^ ones, showcasing the crucial character of intermolecular 1,10-tdapO_2_⋅⋅⋅1,10-tdapO_2_ contacts in the construction of dioxothiadiazole-based magnetic materials [53].

Overall, the dioxothiadiazoles are underexplored when it comes to the design and construction of magnetic materials in combination with paramagnetic metal ions. However, taking into account their rich electrochemistry and chemical tunability, as well as the possibility to form stable radical anions, the library of new switchable compounds is expected to be expanded extensively in the near future.

## 7. Other Properties and Applications

### 7.1. Organic Semiconductors

As a result of their rich electrochemistry, dioxothiadiazoles are considered as good candidates for the construction of *n*-type organic semiconductors [21,51,60]. The chemistry and physical properties of molecules with this particular heterocycle are most commonly discussed in light of electrochemical features: low-lying HOMO and LUMO energy levels, making them good candidates for the *n*-type electron-transporting organic materials. Small organic molecules acting as *n*-type semiconductors are of special research interest as they are necessary for the preparation of purely organic electronics and are not studied as extensively as their *p*-type counterparts [21,60]. The rigid and flat heterocyclic ring of dioxothiadiazoles might also improve the mechanical properties of the designed materials and their overall crystallinity.

### 7.2. Medical and Other Studies

Several different molecules containing dioxothiadiazole moieties have been used in biochemical research as ligands for specific biological targets, for example: CXC- and CC-chemokine receptors in cancer treatment [96,97], ^11^C-labeled radiotracer for selective norepinephrine transporter imaging in PET of cardiac sympathetic nerve system evaluation in patients with heart failure or Parkinson’s disease [98], or histamine H_2_-receptor agonists [62,78,99].

Due to the electrochemical activity of dioxothiadiazoles, the phenanthro[9,10-c]-1,2,5-thiadiazole 1,1-dioxide was tested for its application as a non-toxic corrosion inhibitor for copper in acidic media [100].

## 8. Conclusions and Perspectives

Dioxothiadiazoles present reactivity and properties very different from their parent thiadiazole analogues. The presence of the sulfonyl group strongly influences the electrochemical behavior of these molecules, enabling easy access to paramagnetic radical monoanions. This feature together with the possibility of a straightforward chemical modification at the carbon atoms makes this class of compounds very attractive as potential building blocks for the construction of tuneable functional molecular magnets. The incorporation of dioxothiadiazoles into more complex multinuclear coordination compounds might result in fascinating switchable materials with valence tautomerism, while the polymerization of dioxothiadiazole units might lead to all-organic conductive/magnetic materials. Another research direction might be the merging of multiple dioxothiadiazole groups within a small carbon-based backbone with the aim of reaching organic molecules with multiple valence states. Such molecules would have at least two dublet states (two different *S* = 1/2 states) and possibly a triplet state (*S* = 1 state).

## Figures and Tables

**Figure 1 molecules-26-04873-f001:**
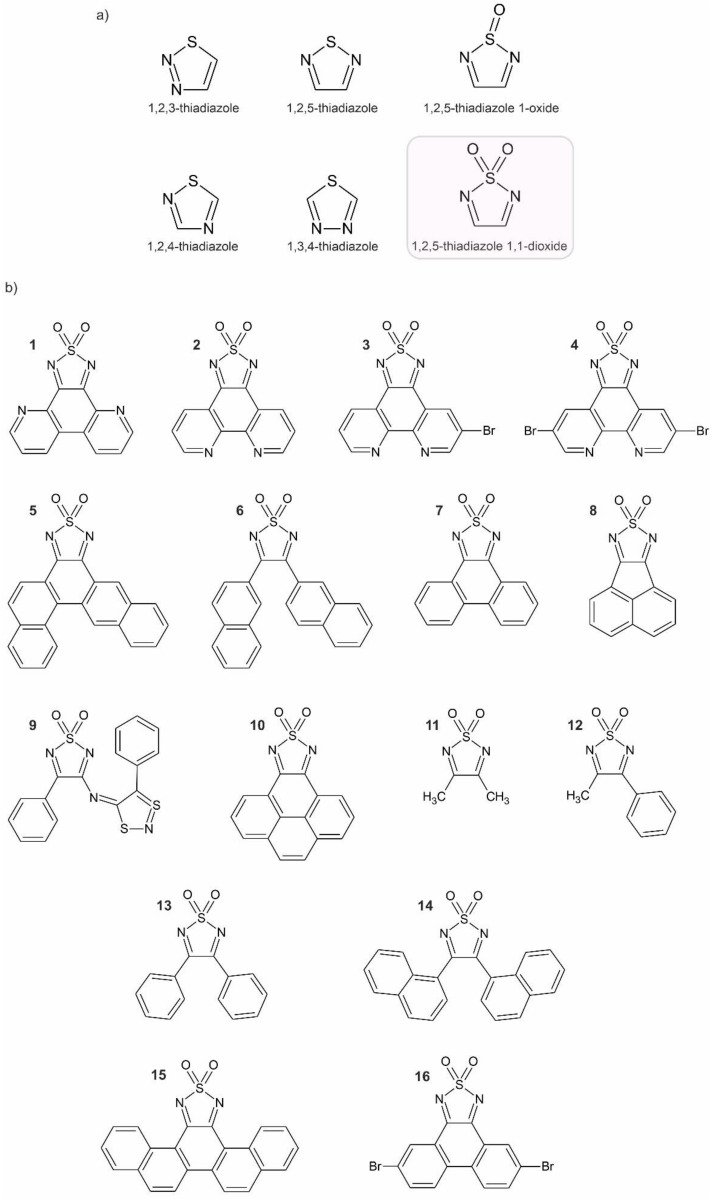
The family of the thiadiazole heterocycles including 1,2,5-thiadiazole 1,1-dioxides (**a**) and the structural formulas of all 1,2,5-thiadiazole 1,1-dioxides characterized structurally by means of SCXRD according to the Cambridge Structural Database search: 10 June 2021 (**b**). See Table 1 for additional details and literature references.

**Figure 2 molecules-26-04873-f002:**
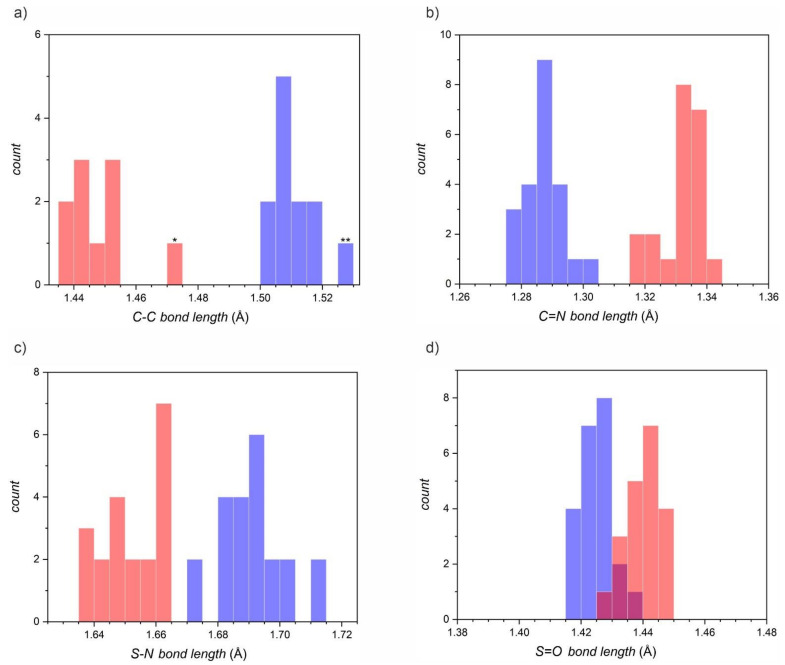
Histograms presenting the distribution of bond lengths in the dioxothiadiazole rings: (**a**) C-C, (**b**) C=N, (**c**) S-N, and (**d**) S=O, depending on the valence state: neutral (blue bars) or radical anion (red bars). Note that each plot has the same bond length axis range of 0.1 Å and the bars correspond to 0.005 Å (outsiders in the C-C distribution: *crystal structure VITSAU, ** crystal structure VITROH).

**Figure 3 molecules-26-04873-f003:**
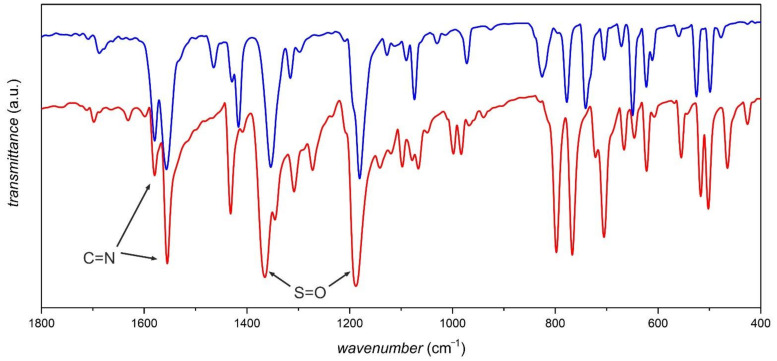
Fingerprint regions of the IR spectra of two dioxothiadiazole derivatives: [1,2,5]thiadiazolo[3,4-f][4,7]phenanthroline 2,2-dioxide (ref. [43]; blue line) and [1,2,5]thiadiazolo[3,4-f][1,10]phenanthroline 2,2-dioxide (ref. [44]; red line) with the C=N and S=O stretching vibration bands indicated.

**Figure 4 molecules-26-04873-f004:**
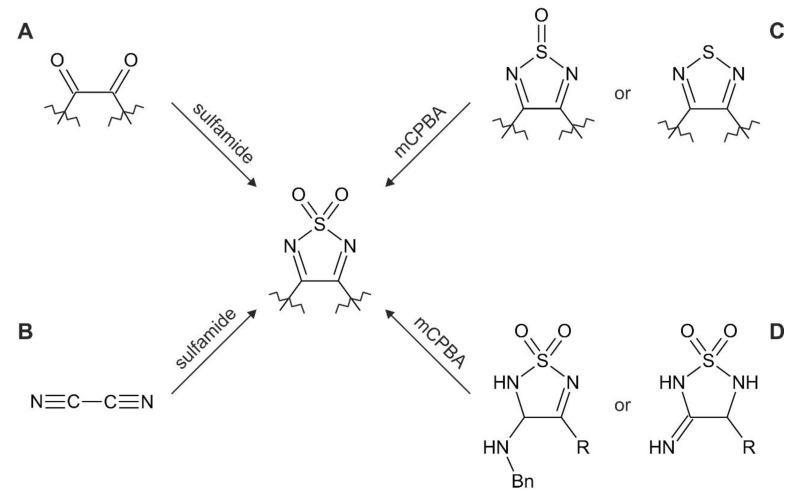
Summary of the synthetic strategies toward 1,2,5-thiadiazole 1,1-dioxides: condensation of 1,2-diketones (**A**) or cyanogen (**B**) with sulfamide or oxidation of the pre-existing heterocyclic scaffolds of 1,2,5-thiadiazoles or 1,2,5-thiadiazole 1-oxides (**C**) or 1,2,5-thiadiazolidines 1,1-dioxides (**D**) using mCPBA.

**Figure 5 molecules-26-04873-f005:**
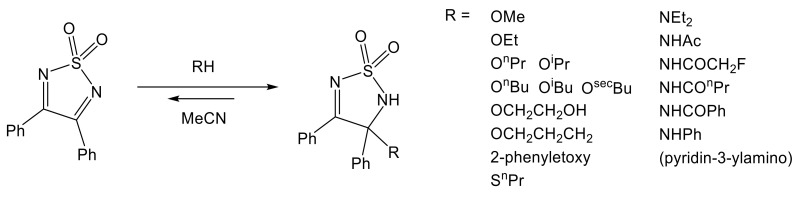
Addition of soft nucleophiles (primary and secondary alcohols, amines, and primary amides) to the C=N double bond of 3,4-diphenyl 1,2,5-thiadiazole 1,1-dioxide [68,82,83,84,85,86].

**Figure 6 molecules-26-04873-f006:**
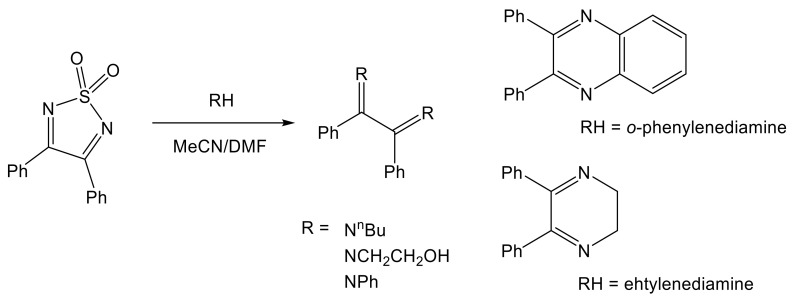
Double addition of soft nucleophiles to C=N bonds in 3,4-diphenyl 1,2,5-thiadiazole 1,1-dioxide followed by the heterocyclic ring cleavage [85,87].

**Figure 7 molecules-26-04873-f007:**
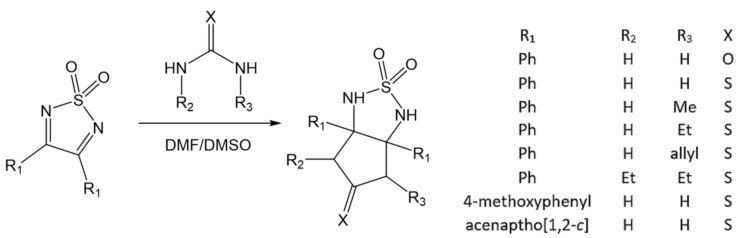
Double addition of substituted ureas/tioureas to C=N bonds of 3,4-diphenyl 1,2,5-thiadiazole 1,1-dioxide [73,86].

**Figure 8 molecules-26-04873-f008:**
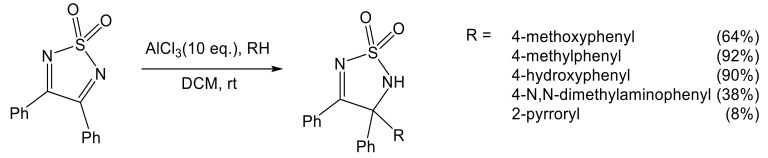
AlCl_3_ catalyzed addition of phenols to the C=N double bond of 3,4-diphenyl 1,2,5-thiadiazole 1,1-dioxide [46,91].

**Figure 9 molecules-26-04873-f009:**
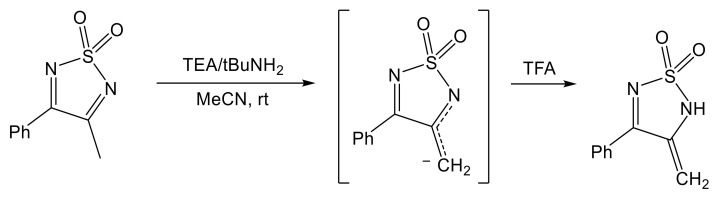
Abstraction of proton from methyl group of 3-methyl-4-phenyl 1,2,5-thiadiazole 1,1-dioxide with the formation of tautomeric carbanion followed by the separation of a 3-methylene-4-phenyl-2,3-dihydro-1,2,5-thiadiazole 1,1-dioxide by the addition of strong acid [93].

**Figure 10 molecules-26-04873-f010:**
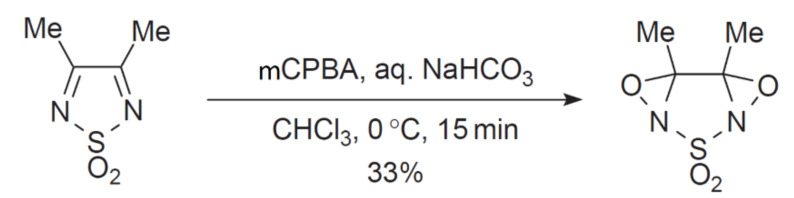
Oxidation of C=N bonds in 3,4-dimethyl 1,2,5-thiadiazole 1,1-dioxide with mCPBA leading to bis-oxaziridine [3].

**Figure 11 molecules-26-04873-f011:**
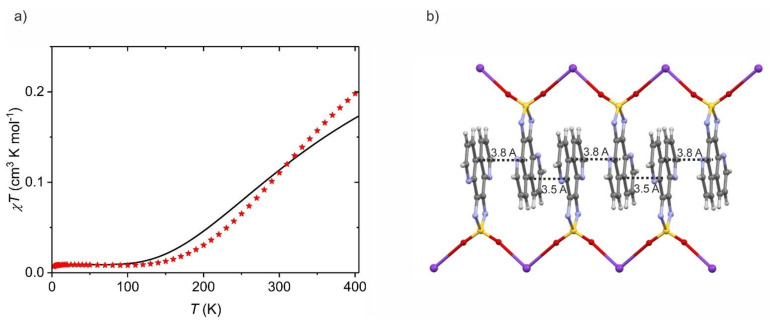
Typical *χT*(*T*) dependence (red stars) recorded at 1000 Oe for K^+^[4,7-tdapO_2_]^−^ [42] (**a**) showing the importance of the π-π stacking interactions (**b**) as efficient magnetic interaction pathways. The solid line in (**a**) is a simulation assuming local antiferromagnetic interactions between two 4,7-tdapO_2_^−^ radical anions (*g* = 2.00) with *J* = −250 cm^−1^ (*H = −2 JS_1_S_2_*).

**Figure 12 molecules-26-04873-f012:**
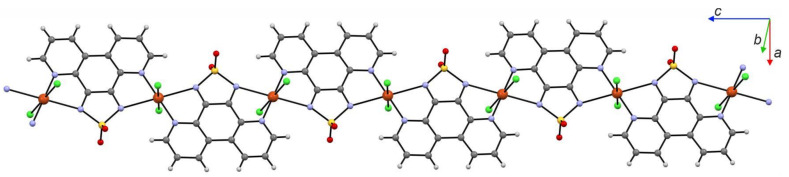
Schematic structure of {CuCl_2_(4,7-tdapO_2_)}_n_ chain with 4,7-tdapO_2_ molecules acting as diamagnetic bridges between the Cu^II^ metal ions [42].

**Figure 13 molecules-26-04873-f013:**
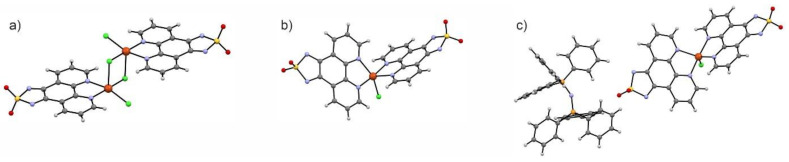
A family of Cu^II^-1,10-tdapO_2_ complexes where the 1,10-tdapO_2_ ligands exhibit different valence states: two neutral ligands in a Cu^II^_2_ dimer (**a**), Cu^II^ complex with one neutral and one radical anion (**b**) and a similar Cu^II^ complex with two radical anions (**c**) [53].

**Table 1 molecules-26-04873-t001:** All 1,2,5-thiadiazole 1,1-dioxides characterized structurally by means of the single crystal X-ray diffraction (according to the Cambridge Structural Database search using WebCSD tool: 10th June 2021).

No. in Figure 1b	Compound Name	CCDC Number (Database ID)	Ref.
1	[1,2,5]thiadiazolo[3,4-f][4,7]phenanthroline 2,2-dioxide (4,7-tdapO2)	997100 (BOKXEG)	[43]
2	[1,2,5]thiadiazolo[3,4-f][1,10]phenanthroline 2,2-dioxide (1,10-tdapO2)	819976 (NALXOP)	[44] [14]
		932270 (MINREI)	
3	5-bromo-[1,2,5]thiadiazolo[3,4-f][1,10]phenanthroline 2,2-dioxide	1882326 (FOFFAK)	[45]
4	5,10-dibromo-[1,2,5]thiadiazolo[3,4-f][1,10]phenanthroline 2,2-dioxide	1882327 (FOFFEO)	[45]
5	benzo[1,2]tetrapheno[5,6-c][1,2,5]thiadiazole 8,8-dioxide	267853 (GESRUS)	[46]
6	3,4-di(naphthalen-2-yl)-1,2,5-thiadiazole 1,1-dioxide	267854 (GESSAZ)	[46]
7	phenanthro[9,10-c][1,2,5]thiadiazole 2,2-dioxide	1179881 (IGISIA) 783889 (IGISIA01) 1577093 (IGISIA02)	[47] [21] [48]
8	acenaphtho[1,2-c][1,2,5]thiadiazole 8,8-dioxide	1179882 (IGISOG)	[47]
9	N-(1,1-dioxo-4-phenyl-1,2,5-thiadiazol-3-yl)-4′-phenyl-1′,3′,2′-dithiazole-5′-imine	1192789 (KAMYEC)	[49]
10	pyreno[4,5-c][1,2,5]thiadiazole 10,10-dioxide	783892 (ONIPEH)	[21]
11	3,4-dimethyl-1,2,5-thiadiazole 1,1-dioxide	1231828 (PEXMEK)	[50]
12	3-methyl-4-phenyl-1,2,5-thiadiazole 1,1-dioxide	1231842 (PEXQAK)	[50]
13	3,4-diphenyl-1,2,5-thiadiazole 1,1-dioxide	1231843 (PEXQEO)	[50]
14	3,4-di(naphthalen-1-yl)-1,2,5-thiadiazole 1,1-dioxide	972953 (VITRIB)	[51]
15	piceno[13,14-c][1,2,5]thiadiazole 14,14-dioxide	972954 (VITROH)	[51]
16	5,10-dibromophenanthro[9,10-c][1,2,5]thiadiazole 2,2-dioxide	1577094 (ZIMLUF)	[48]

**Table 2 molecules-26-04873-t002:** Selected bond lengths for neutral and anionic dioxothiadiazoles (structures collected at 100–173 K). The average values given in blue and red correspond to the blue and red bars in Figure 2.

Compound	S=O (Å)	S-N (Å)	C=N (Å)	C-C (Å)	CSD ID	*T* (K)
[1,2,5]thiadiazolo[3,4-f][1,10]phenanthroline 2,2-dioxide	1.429	1.687	1.287	1.508	NALXOP	173
	1.424	1.686	1.290			
	1.425	1.684	1.290	1.507	MINREI	173
	1.427	1.690	1.285	1.510	MINROS	173
	1.421	1.684	1.286			
	1.425	1.685	1.282	1.515	MINRIM	173
[CuIICl(1,10-tdapO2)](μ-Cl)2[CuIICl(1,10-tdapO2)]	1.419	1.703	1.273	1.503	DUZCIN	120
	1.419	1.682	1.287			
5-bromo-[1,2,5]thiadiazolo[3,4-f][1,10]phenanthroline 2,2-dioxide	1.423	1.694	1.293	1.506	FOFFAK	120
	1.429	1.693	1.285			
5,10-dibromo-[1,2,5]thiadiazolo[3,4-f][1,10]phenanthroline 2,2-dioxide	1.422	1.693	1.290	1.514	FOFFEO	161
	1.425	1.696	1.284			
[1,2,5]thiadiazolo[3,4-f][4,7]phenanthroline 2,2-dioxide	1.430	1.700	1.291	1.519	BOKXEG	100
	1.427	1.699	1.287			
pyreno[4,5-c][1,2,5]thiadiazole 10,10-dioxide	1.417	1.691	1.295	1.518	ONIPEH	173
	1.434	1.683	1.291			
phenanthro[9,10-c][1,2,5]thiadiazole 2,2-dioxide	1.432	1.686	1.288	1.510	IGISIA01	173
	1.417	1.691	1.294			
piceno[13,14-c][1,2,5]thiadiazole 14,14-dioxide	1.438	1.670	1.302	1.529	VITROH	123
	1.427	1.674	1.300			
[CuIICl(4,7-tdapO2)CuIICl]	1.428	1.713	1.275	1.503	BOKXIK	100
	1.428	1.713	1.270			
**Average in neutral species ^1^**	**1.** **426(3)**	**1.** **691(7)**	**1.** **287(5)**	**1.** **512(7)**	**-**	**-**
PPN^+^[Cu^II^Cl(1,10-tdapO_2_^−^)_2_]	1.439	1.662	1.331	1.450	DUZCUZ	120
	1.433	1.642	1.325			
	1.431	1.660	1.320	1.453		
	1.434	1.664	1.321			
PPN^+^[1,10-tdapO_2_]^−^	1.439	1.646	1.333	1.446	FOFFIS	120
	1.442	1.654	1.334			
	1.442	1.664	1.343	1.443	FOFFOY	120
	1.448	1.664	1.340			
PPN^+^[5,10-diBr-1,10-tdapO_2_]^−^	1.443	1.646	1.333	1.452	FOFFUE	117
	1.443	1.661	1.336			
	1.442	1.660	1.337	1.435		
	1.445	1.656	1.333			
PPN^+^[5-Br-1,10-tdapO_2_]^−^	1.437	1.648	1.334	1.441	FOFGAL	120
	1.432	1.659	1.332			
K^+^[1,10-tdapO_2_]^−^	1.449	1.646	1.338	1.441	MINQEH	173
	1.448	1.639	1.337			
NEt^+^[ptdaO_2_]^−^	1.442	1.640	1.340	1.471	VITSAU	173
	1.447	1.636	1.337			
K^+^[4,7-tdapO_2_]^−^	1.438	1.638	1.330	1.437	VONZOP	110
	1.437	1.651	1.320			
**Average in radical anions ^1^**	**1.440(** **4)**	**1.** **652(7)**	**1.** **333(4)**	**1.** **447(10)**	**-**	**-**
**Relative difference between neutral and radical**	**1.0(0.3)%**	**2.3(0.4)%**	**3.5(0.3)%**	**4.3(0.7)%**	**-**	**-**

^1^ ± 3σ_mean_, where σ_mean_—standard deviation of the mean given with formula $\sigma_{mean}=\frac{\sigma }{\sqrt{n}}$, σ is the standard deviation and *n* is the number of observations. 1,10-tdapO_2_-[1,2,5]thiadiazolo[3,4-f][1,10]phenanthroline 2,2-dioxide; ptdaO2-piceno[13,14-c][1,2,5]thiadiazole 14,14-dioxide; 4,7-tdapO2-[1,2,5]thiadiazolo[3,4-f][4,7]phenanthroline 2,2-dioxide; PPN^+^-bis(triphenylphosphine)iminium cation.

**Table 3 molecules-26-04873-t003:** Summary of the synthetic strategies toward 1,2,5-thiadiazole 1,1-dioxides as depicted in Figure 4.

Synthesis Type According to Figure 2	Synthesis Details	Product	Ref.
A	ethanol, reflux, 12–168 h	[1,2,5]thiadiazolo[3,4-f][1,10]phenanthroline 2,2-dioxide [1,2,5]thiadiazolo[3,4-f][4,7]phenanthroline 2,2-dioxide 5-bromo-[1,2,5]thiadiazolo[3,4-f][1,10]phenanthroline 2,2-dioxide 5,10-dibromo-[1,2,5]thiadiazolo[3,4-f][1,10]phenanthroline 2,2-dioxide	[44] [43] [45]
A	ethanol, dry HCl, reflux, 2–3 h	3,4-diphenyl-1,2,5-thiadiazole 1,1-dioxide, acenaphtho[1,2-c][1,2,5]thiadiazole 8,8-dioxide	[75]
A	methanol, MeONa, 80 °C, overnight	sodium 4-phenyl-1,2,5-thiadiazol-3-olate 1,1-dioxide disodium 1,2,5-thiadiazole-3,4-bis(olate) 1,1-dioxide	[76] [72]
A	NEt_3_, MW (360 W), 10 min	3,4-diphenyl-1,2,5-thiadiazole 1,1-dioxide	[77]
A	solvent-free, molybdophosphoric acid (MPA), rt-150 °C, 3–530 h	3,4-diphenyl-1,2,5-thiadiazole 1,1-dioxide, 3-methyl-4-phenyl-1,2,5-thiadiazole 1,1-dioxide, 3,4-dimethyl-1,2,5-thiadiazole 1,1-dioxide, phenanthro[9,10-c][1,2,5]thiadiazole 2,2-dioxide, pyreno[4,5-c][1,2,5]thiadiazole 10,10-dioxide, 3,4-di(naphthalen-2-yl)-1,2,5-thiadiazole 1,1-dioxide, 3,4-di([1,1′-biphenyl]-4-yl)-1,2,5-thiadiazole 1,1-dioxide, 3,4-bis(4-chlorophenyl)-1,2,5-thiadiazole 1,1-dioxide	[74]
A	N,N’-bis(TMS)sulfamide, toluene, BF_3_·Et_2_O, rt, 8 h	3,4-diphenyl-1,2,5-thiadiazole 1,1-dioxide, 3-methyl-4-phenyl-1,2,5-thiadiazole 1,1-dioxide, 3-ethyl-4-phenyl-1,2,5-thiadiazole 1,1-dioxide, 3-ethyl-4-(4-methoxyphenyl)-1,2,5-thiadiazole-1,1-dioxide, 3-Isopropyl-4-(4-methoxyphenyl)-1,2,5-thiadiazole-1,1-dioxide, 3-ethyl-4-(4-chlorophenyl)-1,2,5-thiadiazole-1,1-dioxide, 3-ethyl-4-(furan-2-yl)-1,2,5-thiadiazole-1,1-dioxide, 3-ethyl-4-(thiophen-3-yl)-1,2,5-thiadiazole-1,1-dioxide	[64]
B	diglyme, −60 °C, dry HCl	3,4-diamino-1,2,5-thiadiazole 1,1-dioxide	[63]
C	DCM, mCPBA, reflux, overnight	phenanthro[9,10-c][1,2,5]thiadiazole 2,2-dioxide, pyreno[4,5-c][1,2,5]thiadiazole 10,10-dioxide, 3,4-dimethoxy-1,2,5-thiadiazole 1,1-dioxide	[21] [78]
D	THF, KO_2_ or DMF, NaOEt	3-amino-4-phenyl-1,2,5-thiadiazole 1,1-dioxide, 3-amino-4-(4-methoxyphenyl)-1,2,5-thiadiazole 1,1-dioxide, 3-amino-4-(naphthalen-1-yl)-1,2,5-thiadiazole 1,1-dioxide, 3-amino-4-hexyl-1,2,5-thiadiazole 1,1-dioxide, 3-(benzylamino)-4-phenyl-1,2,5-thiadiazole 1,1-dioxide, 3-(benzylamino)-4-(4-methoxyphenyl)-1,2,5-thiadiazole 1,1-dioxide	[79] [67]

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
