# Peer review of "1,2,5-Thiadiazole 1,1-dioxides and Their Radical Anions: Structure, Properties, Reactivity, and Potential Use in the Construction of Functional Molecular Materials"

_molecules, 2021, doi:10.3390/molecules26164873_

Round 1

Reviewer 1 Report

The review entitled "1,2,5-thiadiazole 1,1-dioxides and their radical anions: structure, properties, reactivity and potential use in the construction of functional molecular materials" deals about the synthesis and the functionnalization of the 1,2,5-thiadiazole 1,1-dioxides cores, and potential use. 

The article is well written and deals about some achievement in this fields.

The article suits well for Molecules and it is ready for publication. 

Only one point before publication. p 7/22, there are some structures (1,10-tdapO2) that are also present in the Figure 1. Maybe the authors can add the abbreviation of the structure p 7 in the figure 1. 

with theses changes the article can be published.

Author Response

We are grateful to the reviewer 1 for comments and suggestions. Below is our point by point response and the description of the changes.

Only one point before publication. p 7/22, there are some structures (1,10-tdapO2) that are also present in the Figure 1. Maybe the authors can add the abbreviation of the structure p 7 in the figure 1

The acronym  1,10-tdapO2 is mentioned in the text in page 7 and is preceded by a full chemical name of this compound. The full name is also used in Tables 1 and 2 together with the CSD database ID and references to relevant publications. Adding this acronym in Figure 1 would be inconsistent with the lack of acronyms for other presented structures. We believe that the numbering scheme provided in Figure 1 and Table 1 is sufficient for the easy identification of 1,10-tdapO2.

Reviewer 2 Report

Pakulski and Pinkowicz submitted a review on the chemical structure, characterization, and functional properties of 1,2,5 thiadizole 1,1 dioxides. 

The review is well organized and relatively compact. Therefore it is easy to read also for students. I have just some minor comments on how the contents could be reorganized to make the text even more intuitive. For example, table 2 is very long and essentially contains three bond lengths. Maybe three histogram plots could be more effective in the main text (the table could be moved in SI). In this way it could be easier to have an average length and also to spot outsiders in the distribution, that might be of interest for several applications. 

In section 2.2 a typical IR spectrum could help the reader in identifying the shape of the peaks that correspond to the typical stretching vibrations of the molecule.

In section 5.1 the authors state that the presence of electron withdrawing groups affect the g factor of the molecule. A graph could help. In section 5.2 a graph of a remarkably important example of coupling could be reported (e.g. ChiT vs T).

The english language is a bit hard to read sometimes. In particular, the authors should pay attention to repetitions and the excessive use of adverbs (extremely, surprisingly, interestingly...). Lines 109-117 contain several "Moreover". In lines 382-386 there are two "extremely". 

Author Response

The response letter is attached
